# Urban rats as carriers of invasive *Salmonella* Typhimurium sequence type 313, Kisangani, Democratic Republic of Congo

Dadi Falay[1,2,3], Liselotte Hardy [3]*, Jacques Tanzito[4], Octavie Lunguya[5,6], Edmonde Bonebe[5], Marjan Peeters[3], Wesley Mattheus[7], Chris Van Geet[8], Erik Verheyen[9,10], Dudu Akaibe[4], Pionus Katuala[4], Dauly Ngbonda[1], François-Xavier Weill[11], Maria Pardos de la Gandara[11], Jan Jacobs[2,3]

**1** Department of Pediatrics, University Hospital of Kisangani, Kisangani, the Democratic Republic of the Congo, **2** Department of Microbiology, Immunology and Transplantation, KU Leuven, Leuven, Belgium, **3** Department of Clinical Sciences, Institute of Tropical Medicine, Antwerp, Belgium, **4** Biodiversity Monitoring Center (Centre de Surveillance de la Biodiversité, CSB), Faculty of Science, University of Kisangani, Kisangani, the Democratic Republic of the Congo, **5** Department of Medical Biology, National Institute for Biomedical Research, Kinshasa, the Democratic Republic of the Congo, **6** Department of Microbiology, University Teaching Hospital of Kinshasa, Kinshasa, Democratic Republic of the Congo, **7** Sciensano, Infectious Diseases in Humans, Bacterial Diseases, Brussels, Belgium, **8** Department of Cardiovascular Sciences and Pediatrics, KU Leuven and University Hospital Leuven, Leuven, Belgium, **9** OD Taxonomy & Phylogeny, Royal Belgian Institute of Natural Sciences, Brussels, Belgium, **10** Evolutionary Ecology, University of Antwerp, Antwerp, Belgium, **11** Institut Pasteur, Université Paris Cité, Unité des bactéries pathogènes entériques, Paris, France

* lhardy@itg.be

**Data Availability Statement:** All relevant data are within the manuscript and its Supporting Information files. All genomes obtained by WGS

## Abstract

### Background

Invasive non-typhoidal *Salmonella* (iNTS–mainly serotypes Enteritidis and Typhimurium) are major causes of bloodstream infections in children in sub-Saharan Africa, but their reservoir remains unknown. We assessed iNTS carriage in rats in an urban setting endemic for iNTS carriage and compared genetic profiles of iNTS from rats with those isolated from humans.

### Methodology/Principal findings

From April 2016 to December 2018, rats were trapped in five marketplaces and a slaughterhouse in Kisangani, Democratic Republic of the Congo. After euthanasia, blood, liver, spleen, and rectal content were cultured for *Salmonella*. Genetic relatedness between iNTS from rats and humans—obtained from blood cultures at Kisangani University Hospital—was assessed with multilocus variable-number tandem repeat (VNTR) analysis (MLVA), multilocus sequence typing (MLST) and core-genome MLST (cgMLST). 1650 live-capture traps yielded 566 (34.3%) rats (95.6% *Rattus norvegicus*, 4.4% *Rattus rattus*); 46 (8.1%) of them carried *Salmonella*, of which 13 had more than one serotype. The most common serotypes were II.42:r:- (n = 18 rats), Kapemba (n = 12), Weltevreden and Typhimurium (n = 10, each), and Dublin (n = 8). *Salmonella* Typhimurium belonged to MLST ST19 (n = 7 rats) and the invasive ST313 (n = 3, isolated from deep organs but not from rectal content). Sixteen

and analyzed on EnteroBase are available at the ENA depository under project number PRJEB54047.

**Funding:** JJ received financial support for the study from the Flemish Interuniversity Council (VLIR-UOS, rat sampling) and the Belgian Directorate General of Development Cooperation (blood culture surveillance). XFW received financial support for the microbial genomics analyses from the Institut Pasteur and the French Government's Investissement d'Avenir program, Laboratoire d'Excellence "Integrative Biology of Emerging Infectious Diseases" (grant no. ANR-10-LABX-62-IBEID. D.F. has a PhD scholarship from KU Leuven through the Marc Vervenne fund. The funders had no role in study design, data collection and analysis, decision to publish, or preparation of the manuscript.

**Competing interests:** The authors have declared that no competing interests exist.

human *S.* Typhimurium isolates (all ST313) were available for comparison: MLVA and cgMLST revealed two distinct rat-human clusters involving both six human isolates, respectively, *i.e.* in total 12/16 human ST313 isolates. All ST313 Typhimurium isolates from rats and humans clustered with the ST313 Lineage 2 isolates and most were multidrug resistant; the remaining isolates from rats including *S.* Typhimurium ST19 were pan-susceptible.

## Conclusion

The present study provides evidence of urban rats as potential reservoirs of *S.* Typhimurium ST313 in an iNTS endemic area in sub-Saharan Africa.

## Author summary

Dadi (°1974, DR Congo) is a Medical Doctor (Kisangani University 2005) with a master in pediatrics (Kisangani University 2015) with special interest in infectious diseases and tropical medicine. He has 11 years of field research experience. He was team member of scientific expedition "*Boyekoli Ebale Congo*" in 2010 as medical support for the researchers. He conducted work field in a multidisciplinary framework with biologists from faculty of sciences (university of Kisangani) exploring zoonotic diseases in several places in the Congo. Currently, he is doing his PhD research at KU Leuven and Institute of Tropical Medicine Antwerp (ITM), Belgium. Passionate about transmissible diseases, Dadi is exploring the potential reservoirs of non-typhoidal *Salmonella* in Kisangani, DR Congo.

## 1 Introduction

Worldwide, invasive non-typhoidal *Salmonella* (iNTS) infections account for 535,000 (95% uncertainty interval 409 000–705 000) invasive disease cases. Sub-Saharan Africa accounts for the largest proportion (79.9%), the highest incidence (34.5% (26.6–45.0)) and a fatality ratio of 15.8% (10.0–22.9%) [1]. Risk factors for iNTS infection in sub-Saharan Africa are age < 5 years old, *Plasmodium falciparum* malaria and HIV-infection [2,3]. Main invasive NTS serotypes are *Salmonella enterica* subspecies *enterica* Typhimurium and Enteritidis (hereafter named *S.* Typhimurium and *S.* Enteritidis), in particular *S.* Typhimurium ST313 [4–7]. This contrasts with high-income countries, where most frequent *S.* Typhimurium sequence type is ST19 [8,9], that causes self-limiting diarrhea with only occasional progression to bloodstream infection [10]. In addition to their virulence, iNTS display high prevalence of antimicrobial resistance [2,11,12].

In high-income countries, the reservoir of non-typhoidal *Salmonella* is zoonotic and transmission is mainly foodborne [10,13,14]. In contrast, iNTS infections in sub-Saharan Africa are believed to be anthroponotic [7,15,16]. In line with *Salmonella* Typhi and Paratyphi A causing enteric fever in humans [17], invasive *S.* Typhimurium and Enteritidis have genetically adapted to humans [2,11]. These invasive NTS are clinically specific: they are mainly causing bloodstream infections (and not diarrhea) and are associated with high mortality [18,19]. Further, household studies of index patients with NTS bloodstream infections in Kenya and Burkina-Faso have demonstrated NTS stool excretion in household members but not in household livestock [15,20]. However, reviewing the evidence of source attribution, several

authors stated that an animal reservoir for NTS in sub-Saharan Africa is not excluded and needs further study [21–23].

The Democratic Republic of Congo (DR Congo) is highly endemic for iNTS disease. The National Institute of Biomedical Research (INRB) in Kinshasa coordinates, in collaboration with the Institute of Tropical Medicine (ITM) in Antwerp, Belgium, a nationwide microbiological surveillance network processing blood cultures in patients suspected of bloodstream infections. Kisangani University Hospital participates—through hospitals and health centers within and around Kisangani—in this network [24]. Over the past decade, NTS consistently represented the most frequent blood culture isolates in children attending the hospitals and health centers in this network, with *S.* Typhimurium and Enteritidis isolates representing respectively 63.1% and 36.4% of iNTS isolates of which 87.4% displaying MDR [19].

As a fast-growing densely populated metropolis, Kisangani provides a wide array of ecological niches attractive for rodents. Previously, Kisangani University had assessed rats trapped in marketplaces in Kisangani as reservoirs of *Rickettsia typhi* and *Bartonella* spp. [25]. Rodents have been described as a reservoir and vector of many pathogens including *Salmonella* [26] but to our knowledge, rats have not yet been assessed as a potential reservoir of NTS. Therefore, we aimed to assess rats present in marketplaces in Kisangani for carriage of NTS. Our objectives were to estimate the proportion of *S.* Typhimurium and Enteritidis carriers (rats with *S.* Typhimurium or Enteritidis isolates from liver, spleen, blood, and/or intestinal content) among rats, and to assess genetic relatedness between rat and human NTS isolates. A secondary objective was to assess the antimicrobial resistance profile of the *Salmonella* isolates.

## 2 Materials and methods

### 2.1 Ethics statement

Ethical approval for the blood culture surveillance study was granted by the Institute of Tropical Medicine Antwerp Institutional Review Board (approval number: 613/08), the Ethics committee of Antwerp University (approval number: 8/20/96) and the Ministry of Health of the Democratic Republic of the Congo (approval number: 8/20/96). For the rat carrier study, authorization by the health authorities of Tshopo Province (Division Provincial de la Santé) was obtained on 12 May 2016 and procedures were carried out according to the 2016 Guidelines of the American Society of Mammologists for the use of wild mammals in research and education [27]. Before starting rat trapping, the responsible in charge of security at the trapping site was contacted for permission and surveillance of the traps.

### 2.2 Study design

In an exploratory study, rats trapped alive in marketplaces in the Kisangani area were sampled for *Salmonella* in the microbiological laboratory at the University Hospital of Kisangani. *Salmonella* isolates were tested for serotype and antimicrobial susceptibility. *S.* Typhimurium and Enteritidis isolates were compared by molecular typing for genetic relatedness with clinical isolates obtained from blood cultures.

### 2.3 Rat sampling

**2.3.1 Sample size calculation.**   Based on CLSI guidelines, a minimum sample size of 30 *Salmonella* spp. isolates was set for reliable data about serotype distribution and antimicrobial susceptibility profiles [28]. At an assumed NTS carrier ratio of 5–10% [29–32] we targeted to capture a total of 600 rats to have sample size of 30 rats carrying *Salmonella*.

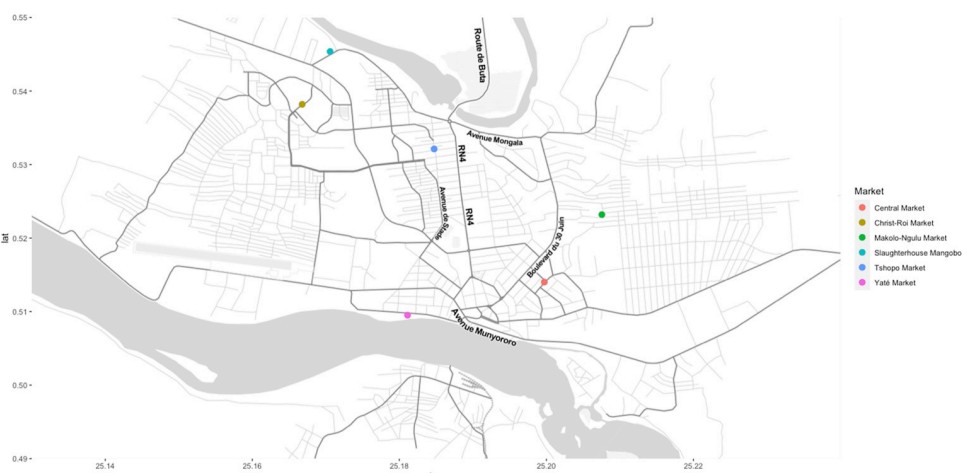

**Fig 1. Overview of sampling sites in Kisangani city.** Kisangani map was made using OpenStreetMap data, Stamen open source maps (https://stamen.com/open-source/) and the R-package ggmap [34].

**2.3.2 Rat collection settings.** The rat sampling was conducted between April 2016 and December 2018, with interruptions due to administrative and/or logistical issues from January to February 2017 and from April 2017 to April 2018 (S1 Fig). Kisangani counts approximately one million inhabitants; 70 to 80% are living below the poverty line [33]. Sampling sites were selected and described previously [25] (Fig 1). The Central Market is the largest market in Kisangani city, located in the city center, as is the case for Makolo-Ngulu market, Tshopo market and Christ-Roi market. Yaté Market is in the secondary port of Kisangani, at the Congo River. Merchants from the surrounding villages temporarily reside at the market. The Mangobo slaughterhouse, at the bank of the Tshopo river, is situated in a savannah habitat, close to a small port trading cattle imported from the eastern provinces (Ituri and Nord-Kivu). In all these sites, the goods are displayed on tables made of unpolished wood allowing easy access by rats. In some places, the products for sale are placed on the ground (*e.g.* rice or corn put to dry). There is no rodent control policy in the city of Kisangani.

**2.3.2 Rat sampling procedure.** Tomahawk live-capture traps (Tomahawk Trap Co., Tomahawk, WI, USA) (dimensions 490/178/173 mm) were used with smoked fish or meat as bait. The traps were installed in warehouses, rice mills, or sheltered places in the sampling sites, to prevent them from being stolen or misused. Traps were installed between 5–7 p.m. and collected the next day between 6–8 a.m. Using a global positioning system (GPS) (Garmin 60 Cx Southampton, Hampshire, UK), the location of each positive trap was recorded.

Trapping success was defined as the proportion of rats caught per number of traps installed in the trapping site. Trapped rats were transported to the Faculty of Sciences in Kisangani. At reception, rats were taken out of the trap using a single-use cloth bag for each animal. Still in the bag (to prevent scratches or bites), they were euthanized with an intraperitoneal injection of a lethal dose of ketamine (Ketalin, Shalina laboratories PVT LTD, Mumbai India, 0.5-1ml of 250mg/5ml solution). The delay between collection from the trapping site and euthanasia was less than 3 hours and the animal was dissected immediately after the euthanasia (less than 10 minutes). Rat species (*Rattus norvegicus* versus *Rattus rattus*) was determined on morphological characteristics. Sexual status, length and mass were recorded.

After euthanasia, rats were placed in a dissection tray and the abdomen was sprayed with 3-5ml of 80% ethanol to disinfect the skin. Using a sterile scalpel blade, an incision was made throughout the length of the abdomen. Different samples (2–3 g, each sampled with separate

sterile instruments) were collected from the spleen and subsequently from the liver. From May 2018 onwards, additional blood (0.5-1ml) was sampled by intracardiac puncture, before sampling of spleen and liver. At the end of the procedure, the rectal content was collected. Internal organs collected were transported to the lab for preliminary analysis within 4 hours.

The samples of spleen, liver and rectal content were placed in a sterile Petri dish, sliced, and transferred into 10ml Selenite broth (Becton Dickinson and Company, Franklin Lakes, NJ, US) followed by vortex mixing. Blood was inoculated directly into the Selenite broth tube. Selenite broth tubes were transported to the microbiology laboratory within 4 hours after inoculation.

## 2.4 Salmonella identification, serotyping, and antibiotic susceptibility testing in rats

Laboratory processing was done as described previously for stool samples [35]. Selenite broth tubes were incubated at 35˚C for 12–18 hours, whereafter 10 μl broth was inoculated on two *Salmonella-Shigella* (SS) agar plates (Lab M Limited, Lancashire, UK). These were incubated at 35˚C and read after 18-24h. In case of no growth, the plates were evaluated after another 24 hours of incubation. In case of growth, up to five (to detect the presence of multiple seroptypes) colonies suspected of *Salmonella* were transferred to Kligler Iron Agar (KIA) tubes (Lab M Limited) and incubated for 18–24 hours at 35˚C. Bacterial growth in the KIA tubes suggestive of *Salmonella* was biochemically confirmed by a panel of disk-based biochemical tests (DiaTabs, Rosco, Taastrup, Denmark). Isolates with a reaction pattern compatible with *Salmonella* were stored in 2 ml tubes of Trypticase Soy Agar (Oxoid, Basingstoke, UK) and shipped to the Institute of Tropical Medicine (ITM, Antwerp, Belgium) for serotyping and antibiotic susceptibility testing (AST).

Serotyping was done by commercial antisera (Vison, Pro-lab Diagnostics Inc., Richmond Hill, Ontario, Canada). AST was done by disk diffusion (Neo-Sensitabs, Rosco, Taastrup, Denmark) and, in the case of azithromycin and ciprofloxacin, by E-test macromethod (bioMérieux, Marcy L'Etoile, France) for assessment of minimal inhibitory concentration values (MIC-values) as previously described [19] (S3 Table). Results were interpreted according to the Clinical and Laboratory Standards Institute (CLSI) M100-S31 criteria [36]. Multidrug resistance (MDR) was defined as combined resistance to amoxicillin, trimethoprim-sulfamethoxazole, and chloramphenicol [12].

## 2.5 Human sampling

*Salmonella* isolates were obtained and stored as part of this INRB-ITM microbiological blood culture surveillance network at Kisangani University Hospital as previously described [24]. Basic clinical and demographic data were recorded. To assess the genetic relatedness between *Salmonella* isolates from rats and humans, blood culture isolates obtained during the rat trapping period and six months before and 6 months after this period (01 October 2015–30 June 2019) were selected. Serotyping and AST of blood culture *Salmonella* isolates were done as described previously [19].

## 2.6 Whole genome sequencing

For all but one *Salmonella* spp. isolate, whole genome sequencing (WGS) and analysis was performed as previously described [37]. Total DNA was extracted from overnight cultures at 37˚C in tryptic soy broth (TSB) (Oxoid) using the MagNAPure 96 system (Roche Diagnostics, Meylan, France). WGS was performed by the genomic platform of the Institut Pasteur, in Paris, France ("Plateforme de microbiologie mutualisée", P2M). The libraries were prepared with the

Nextera XT kit (Illumina) and sequencing was performed with the NextSeq 500 system (Illumina, San Diego, CA, USA), generating 150 bp paired-end reads. The short reads were assembled de novo with SPAdes version 3.6.0.23. Serotype prediction, multilocus sequence typing (MLST), and core genome MLST (cgMLST) were performed with various tools integrated into EnteroBase (https://enterobase.warwick.ac.uk/). The presence and type of antimicrobial resistance genes (ARGs) or ARG-containing structures were determined with ResFinder version 4.1 (https://cge.cbs.dtu.dk/services/ResFinder/) and PlasmidFinder version 1.3 (https://cge.cbs.dtu.dk/services/PlasmidFinder/) on SPAdes assemblies.

**2.6.1 Genetic relatedness between Salmonella isolates from rats and humans.** Genetic relatedness between the iNTS isolates (in case those belonging to *S.* Typhimurium serotype, see Results) was assessed by three different approaches: multilocus variable-number tandem repeat (VNTR) analysis (MLVA), MLST, and cgMLST.

MLVA was performed at Sciensano (Brussels, Belgium) as described previously [35]. MLVA profiles were established based on five loci ordered STTR9-, STTR5-, STTR6-, STTR10-, and STTR3-. Identical *S.* Typhimurium MLVA types (clusters) were defined as isolates with variation in none or one of the rapidly changing loci (STTR5, STTR6, and STTR10) but no variation in the stable loci (STTR3 and STTR9) [38].

For WGS, the 7-genes MLST scheme was applied, based on the sequences of the seven housekeeping genes: *aroC*, *dnaN*, *hemD*, *hisD*, *purE*, *sucA*, *thrA* [39,40]. Hierarchical clustering on cgMLST (HierCC) and minimum spanning trees (MS-trees) was performed on EnteroBase based on the sequences of 3002 loci for *Salmonella* [41]. On the HierCC scheme, HC2850 values allow to deduct the *Salmonella* subspecies (*i.e.* difference of $\leq$ 150 or less alleles from the 3002 loci in the cgMLST scheme); HC200 values provide a similar discrimination as the 7-genes MLST typing ($\leq$ 200 alleles difference between genomes); and HC5 values can be considered a strong indicator of epidemiological relatedness (i.e. if no more than five alleles are different) [41,42].

**2.6.2 Genetic relatedness between isolates from this study and the literature.** We compared some of the isolates of *S.* Typhimurium and *S.* Enteritidis identified in this study with previous studies on iNTS in Africa [7,43] using EnteroBase. This database recovers regularly *Salmonella* genomes from the GenBank Sequence Read Archive (SRA). The "Custom View" utility allows to add metadata for genomes already present on the site. EnteroBase has recovered 564 genomes from the study on *S.* Typhimurium ST313 by Pulford *et al.* [7] and 503 genomes from the study on *S.* Enteritidis ST11 by Feasey *et al.* [43]. A "Custom View" was created to add the information on lineages and plasmid repertoire for the *S.* Typhimurium ST313 from the Pulford *et al.* study [7], that was next compared to the HierCC clustering on cgMLST by EnteroBase [41]. A "Custom View" previously created, including the hierBAPS clade/cluster data described by Feasey *et al.* [37] was used to compare the hierBAPS clustering with the EnteroBase HierCC clustering. A minimum spanning (MS) tree (MStree V2 or GrapeTree) based on the EnteroBase "cgMLST V2 + HierCC V1" scheme was produced for each serotype to estimate the allelic distances between isolates from this study and the genomes from Pulford *et al.* [7] and Feasey *et al.* [43].

For the *S.* Typhimurium ST313 in this study, four plasmids, previously associated with the main lineages circulating in Africa: pSLT-BT (accession: FN432031.1), pBT1 (accession: LS997975.1), pBT2 (accession: LS997976.1), and pBT3 (accession: LS997977.1) were searched, and this information was included in the *S.* Typhimurium ST313 "Custom View".

For the *S.* Enteritidis ST11 in this study, plasmid pSEN-BT (accession: LN879484.1) described by Feasey *et al.* [43] as characteristic of the hierBAPS clade 9, also denominated Central/Eastern African clade was also searched and the information was included in the *S.* Enteritidis ST11 "Custom View".

**Table 1. Numbers of outreach visits, rats caught and *Salmonella* carriers over the six sampling sites in Kisangani 2016–2018.**

| | Central Market | Yaté Market | Slaughter house Mangobo | Christ-Roi Market | Makolo-Ngulu Market | Tshopo Market | Total |
|---|---|---|---|---|---|---|---|
| **Outreach visits** | 14 | 26 | 5 | 2 | 3 | 5 | 55 |
| **Traps installed** | 420 | 780 | 150 | 60 | 90 | 150 | 1650 |
| **Rats caught (% of traps)** | 128 (30.4%) | 380 (48.7%) | 12 (8.0%) | 4 (6.7%) | 16 (17.8%) | 26 (17.3%) | 566 (34.3%) |
| ***Salmonella* carriers (% of rats)** | 12 (9.4%) | 30 (7.9%) | 2 (16.7%) | 1 (25.0%) | 0 (0.0%) | 1 (3.8%) | 46 (8.1%) |

## 2.7 Data management and analysis, definitions

Geographic coordinates, morphometric measurements of rats and microbiological data were entered in an Excel database (Microsoft, Redmond, WA, USA). Rats with at least one isolate confirmed as *Salmonella* were categorized as *Salmonella* carriers. Several rats carried more than one *Salmonella* serotype. Only the first isolate per serotype per rat was used for the analysis of the distribution of serotypes and AST. For the display of MLVA typing results, all *Salmonella* Typhimurium isolates were assessed but only the first isolate per MLVA type per rat was presented. All genomes obtained by WGS and analyzed on EnteroBase are available at the ENA depository under project number PRJEB54047 (S1 Table).

## 3 Results

### 3.1 Rats sampling

During 55 outreach visits, 566 (34.3%) of 1,650 traps yielded 566 rats including 541 (95.6%) *Rattus norvegicus* (Norwegian or brown rat) and 25 (4.4%) *Rattus rattus* (roof rat or black rat). The highest trapping success was seen in Central and Yaté marketplaces (30.4% and 48.7% respectively). There were no incidents (*e.g.* tripping) with the traps reported and apart from one cat, no other non-rat species was captured. A total of 46 (8.1%) rats carried *Salmonella* spp. (Table 1). All captured rats were alert and active, and none showed signs of illness and weakness. Most rats (89.0%, 504/566) were sexually mature, with a male-to-female ratio of 0.91; median (Interquartile Range, IQR) length (without tail) and mass were 22.1 cm (8.3–28.4 cm) and 280 g (30–515 g) respectively; showing no difference between *Salmonella* carriers and non-carriers (S2 Table).

### 3.2 Distribution of serotypes and specimens in rats

The 46 *Salmonella* rat carriers yielded a total of 254 *Salmonella* isolates, of which 253 were available for further analysis, one isolate was not viable after transport. Table 2 lists the *Salmonella* serotypes isolated per body specimen and rat. Thirteen rats yielded multiple serotypes and a total of 73 non-duplicate serotypes isolates were recovered. Most common serotypes were serotype II.42:r:- (found in 18 rats), Kapemba (12 rats), Weltevreden and Typhimurium (10 rats each), and Dublin (found in 8 rats). All *Salmonella* Dublin non-duplicate serotype isolates (n = 11) and all but one *S*. Typhimurium isolates (n = 12 in 10 rats) were obtained exclusively from deep organ samples (spleen or liver). *S*. Enteritidis was not isolated from rats.

### 3.3 Salmonella serotypes obtained from human blood cultures

A total of 40 non-duplicate *Salmonella* isolates were recovered from a total of 2742 blood cultures performed. Two isolates were not viable upon retrieval from shipment and not available for serotyping. Main serotypes comprised Typhimurium (n = 18), and Enteritidis (n = 11).

**Table 2. Serotypes of rats (n = 46) according to specimen.** First isolate per serotype per specimen. A total of 73 single non-duplicate *Salmonella* serotype isolates were recovered from 46 rats; multiple serotypes were observed in 13 rats. Specimens listed were sampled for each rat, except for blood which was sampled for 218/566 (38.5%) rats.

| | Blood | Liver | Spleen | Rectal content | Total | Numbers (%) of rats carrying the serotype |
|---|---|---|---|---|---|---|
| *Salmonella* Dublin | 0 | 5 | 6 | 0 | 11 | 8 (17.3%) |
| *Salmonella* II:42:r:- | 2 | 6 | 2 | 10 | 20 | 18 (39.1%) |
| *Salmonella* Kapemba | 2 | 6 | 6 | 3 | 17 | 12 (26.0%) |
| *Salmonella* Mikawasima | 0 | 0 | 1 | 1 | 2 | 2 (4.3%) |
| *Salmonella* Orion | 0 | 0 | 0 | 1 | 1 | 1 (2.17%) |
| *Salmonella* Typhimurium | 1 | 6 | 4 | 1 | 12 | 10 (21.7%) |
| *Salmonella* Weltevreden | 0 | 5 | 2 | 3 | 10 | 10 (21.7%) |
| Total | 5 | 28 | 21 | 19 | 73 | 46 |

Other serotypes were Typhi (n = 6), Paratyphi C (n = 1), II.42:r:- (n = 1), and monophasic I.6,7:y:- (n = 1). Median age of patients infected with iNTS (n = 34) was 20.5 months (3 days–48 years), 81.8% (n = 27/33) were less than 5 years old. Male-to-female ratio was 0.83.

### 3.4 *S.* Typhimurium isolates

Results of MLVA and MLST typing of *S.* Typhimurium isolates from rats (12 isolates from 10 rats) and humans (18 isolates from 18 patients available for MLVA typing, 16 isolates from 16 patients available for MLST typing) are listed in Table 3. Fig 2A and 2B depicts the clustering of those 32 isolates through a minimum spanning tree on cgMLST.

The 18 human *S.* Typhimurium isolates comprised nine different MLVA types and all 16 sequenced isolates belonged to MLST ST313. Among the 12 *S.* Typhimurium isolates recovered from 10 rats, four isolates (from three rats) belonged to ST313. The rat ST313 isolates represented 6.5% of *Salmonella*-carrying rats and 0.5% of all rats captured. Rat ST313 isolates belonged to three MLVA types. The remaining eight *S.* Typhimurium isolates from rats belonged to ST19 (four MLVA types, seven rats). The three rats carrying *Salmonella* ST313 did not show signs of illness and had length and mass in line with all the other rats in the study (S2 Table).

Two human-rat clusters were identified among the ST313 isolates, involving the three ST313 carrying rats and 12 out of 16 (75.0%) human isolates (Table 3, Fig 2A and 2B). In the first cluster, two rat isolates (Yate Market, October 2018) grouped with six human isolates obtained between May 2016 and September 2018 (28 months) from patients living between in a perimeter of 3.5 to 3.7 km from the marketplace. In the second cluster, two isolates from one rat (Central Market, October 2016) grouped with six human isolates obtained between August 2016 and May 2017 (8 months) from patients living within a perimeter of 1.2 to 2.9 km from the marketplace.

The ST313 *S.* Typhimurium human and rat isolates clustered with the ST313 Lineage 2 isolates described by Pulford *et al.* [7], at less than 50 alleles (by the EnteroBase cgMLST scheme), and at more than 200 alleles to the Lineage 1 and Lineage 3 isolates in their study (Fig 3). As to the plasmid repertoire described by Pulford *et al.*[7], there were two distinct profiles: pBT2 and pBT3 (n = 6 humans and n = 1 rat) and pBT3 alone (n = 7 humans and n = 2 rats) (Fig 4). The ten ST19 *S.* Typhimurium isolates (all eight rats captured at Yaté and Central markets) clustered both by MLVA and HC5 (Fig 5).

### 3.5 *Salmonella* Enteritidis isolates

Among 11 human *S.* Enteritidis isolates available, 10 (90.9%) belonged to ST11. When compared on EnteroBase with the genomes available from the study Feasey *et al.* [43], they

**Table 3. MLVA types of *Salmonella* Typhimurium isolated from rats and human blood cultures.** NA corresponds to a locus on which no allele has been amplified. The two groups of grey shaded cells present MLVA clusters of rat (bold) and human isolates. Two rats (MC071 and MC025) had each two different MLVA types. Code numbers of rats refer to sampling site: AB = Slaughterhouse (Abattoir) Mangobo, MC = Marché Central, MY = Marché Yaté; and specimen type: S = spleen, L = liver, B = blood, ND = no data.

| MLVA type | Rats | | | Humans | | | |
|---|---|---|---|---|---|---|---|
| | Code number | ST | Date | Code number | ST | Date | Residence township (distance from rat trapping site) |
| 2-5-8-8-210 | - | | - | 5548/4 | 313 | 28/10/16 | Mangobo (3.5 km) |
| 2-5-9-8-210 | **MY302/S** | 313 | 08/10/18 | - | - | - | |
| | **MY305/L** | 313 | 08/10/18 | - | - | - | |
| 2-5-11-8-210 | - | - | - | 5232/4 | 313 | 22/05/16 | Kabondo (3.7 km) |
| | - | - | - | 6284/4 | 313 | 07/08/18 | Kabondo (3.7 km) |
| | - | - | - | 6436/4 | 313 | 06/09/18 | Tshopo (3.5 km) |
| 2-6-11-8-210 | - | - | - | 5248/4 | 313 | 26/08/16 | Unknown |
| | - | - | - | 5376/4 | 313 | 10/08/16 | Unknown |
| 2-8-10-8-210 | - | - | - | 5809/4 | 313 | 13/05/17 | Kisangani (2.9 km) |
| | - | - | - | 5810/4 | 313 | 13/05/17 | Kisangani (2.9 km) |
| 2-7-10-8-210 | **MC071/L+S** | 313 | 06/10/16 | 5403/4 | 313 | 27/08/16 | Kabondo (2.6 km) |
| | | | | 5464/4 | 313 | 23/09/16 | Unknown |
| | | | | 5598/4 | 313 | 22/11/16 | Kisangani (1.2 km) |
| | | | | 5601/4 | 313 | 24/11/16 | Kabondo (2.6 km) |
| 2-7-11-8-210 | **MC071/S** | 313 | 06/10/16 | - | | - | |
| 2-4-11-7-210 | - | - | - | 7281/4 | Not done | 13/06/19 | |
| 2-7-12-10-210 | - | - | - | 4930/4 | Not done | 30/10/15 | |
| 2-13-4-3-NA | - | - | - | 5835/4 | 313 | 31/05/17 | |
| 2-NA-12-7-210 | - | - | - | 5390/4 | 313 | 16/08/16 | |
| | - | - | - | 7279/4 | 313 | 10/06/19 | |
| | - | - | - | 7292/4 | 313 | 14/06/19 | |
| 3-12-5-9-311 | MC025/L+S | 19 | 15/05/16 | - | - | - | |
| 3-12-5-10-311 | MC025/L+S | 19 | 15/05/16 | - | - | - | |
| 3-15-5-11-311 | MY213/S | 19 | 10/06/18 | - | - | - | |
| | MY218/L | 19 | 10/06/18 | - | - | - | |
| | MY365/S | 19 | 16/12/18 | - | - | - | |
| | MY373/L | 19 | 23/12/18 | - | - | - | |
| | MY375/B | 19 | 23/12/18 | - | - | - | |
| 2-20-9-7-212 | MY300/L | 19 | 08/10/18 | - | - | - | |

clustered at less than 50 alleles to the hierBAPS clade 9, also denominated Central/Eastern African clade (S2 Fig), and carried the expected plasmid pSENT-BT.

### 3.6 Antimicrobial resistance profile

*S.* Typhimurium from both rats (all 3) and humans (14 out of 18) displayed MDR, encoding for resistances to a variable number of families of antibiotics. The most frequent profile consisted of AMR determinants to beta-lactams (*bla*$_{TEM-1}$), aminoglycosides (*strA*, *strB*, *ant(3')-Ia*, *aac(3')-IId*), sulfonamides (*sul1*, *sul2*), trimethoprim (*dfrA1*), tetracyclines (*tetB*), chloramphenicol (*catA10*). Among human isolates, occasional decreased ciprofloxacin susceptibility and resistance to azithromycin and ceftriaxone were also observed (Table 4, S3 Table). In contrast, *S.* Typhimurium ST19 isolates from rats and all other serotypes from rats were susceptible to all antibiotics tested.

ST11 *S.* Enteritidis isolates, all recovered from human samples, encoded for AMR determinants to 5 to 6 families of antibiotics: beta-lactams (*bla*$_{TEM-1}$), aminoglycosides (*strA*, *strB*), chloramphenicol (*cat2*), sulfonamides (*sul2*), trimethoprim (*dfrA7*) and tetracyclines (*tetA*).

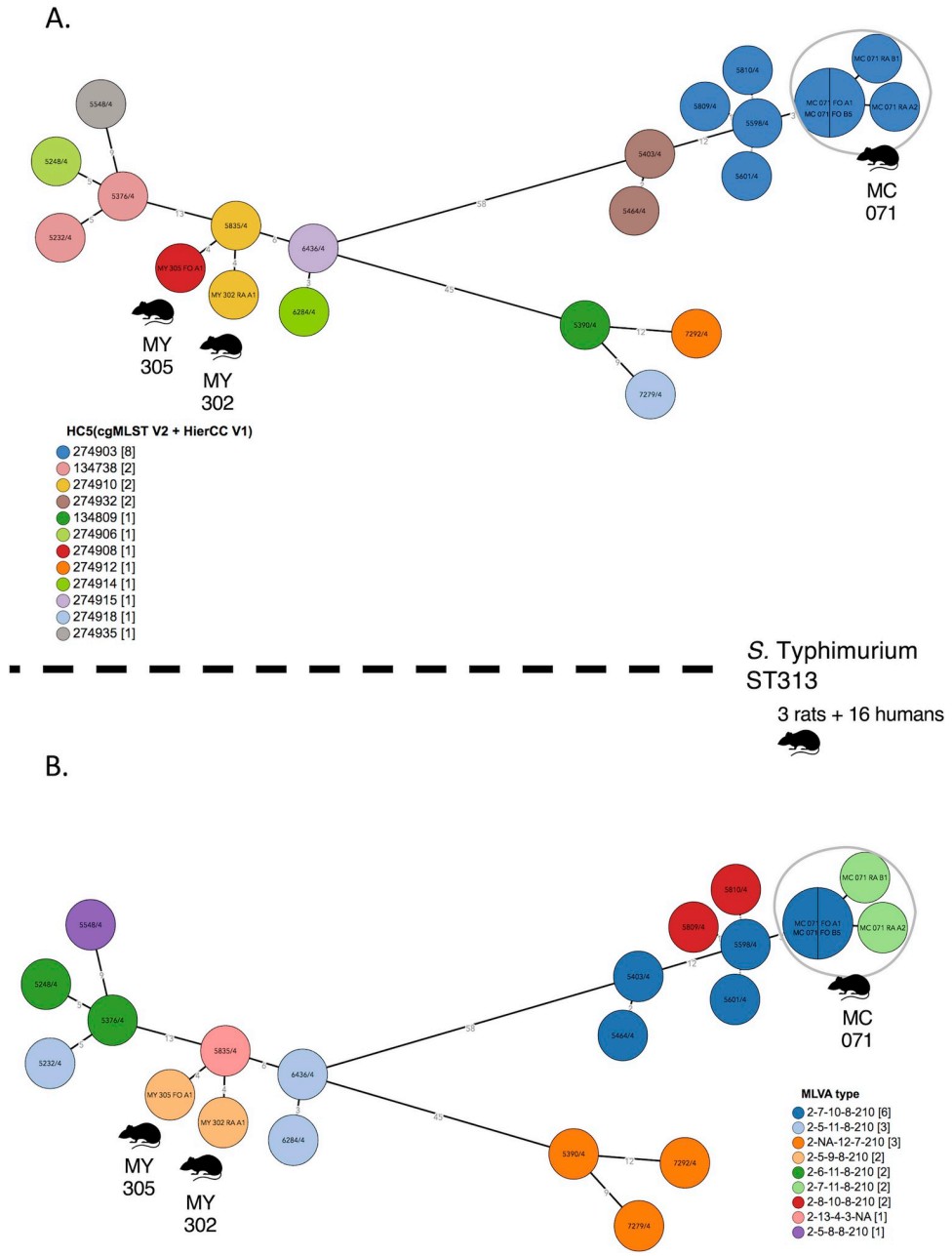

**Fig 2. Clustering of *S.* Typhimurium ST313 human and rat isolates.** Minimum spanning tree created using the MSTree V2 component in EnteroBase, based on the allelic differences over the 3002 alleles that constitute the EnteroBase HierCC scheme on cgMLST [41]. The distances between leaves in the tree indicate number of alleles different between genomes. Genomes with common HC5 values are at five or less alleles, strongly indicating epidemic relatedness. Isolates from rats are indicated with an icon and prefix MC and MY (referring to Christ market and Yate market respectively. In panel A, colors are according to cgMLST HC5 values; in panel B, colors are according to MLVA type. Human and rat isolates are grouping in two clusters (MY305/302 and MC071 with each six human isolates. For details of date and place, see Table 3. Rat drawing by Francisca Arévalo from NounProject.com.

## 4 Discussion

In a sub-Saharan Africa setting endemic for iNTS, the present study showed a *Salmonella* carrier ratio of 8.1% among rats captured in an urban area. Among the serotypes recovered, there

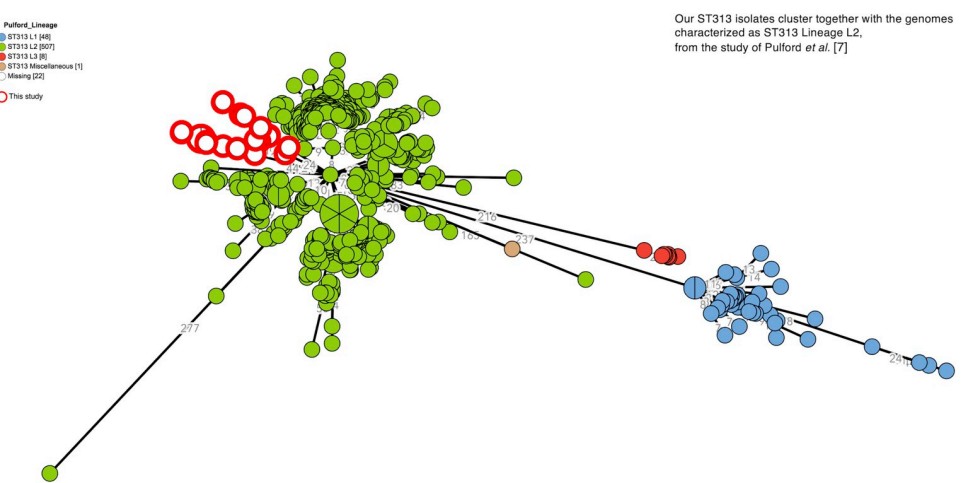

**Fig 3. Clustering of *S.* Typhimurium ST313 human and rat isolates with ST313 Lineage 2 isolates described by Pulford *et al.* [7].** Minimum spanning tree (MSTree V2) comparing the *S.* Typhimurium ST313 genomes analyzed by Pulford et al. [7] in 2021 and the genomes in this study. The ST313 genomes in this study (circled in red, filled in white) cluster together with the Lineage 2 strains in the study by Pulford *et al.* [7], colored in green).

was invasive *S.* Typhimurium ST313 in 3/46 *Salmonella*-carrying rats which clustered with 13/16 human isolates.

## 4.1 Rats as reservoirs of non-Typhoidal Salmonella

Rodents–in particular rats–are well-known reservoirs of zoonotic pathogens including *Salmonella* [26]. Prevalence of *Salmonella* in rats has mainly been assessed in high-income countries and varied widely: in the natural environment, prevalence was consistently low (0 to 1%) [26,44,45] but is was higher in the proximity of human settlements (5–20%) such as cities and

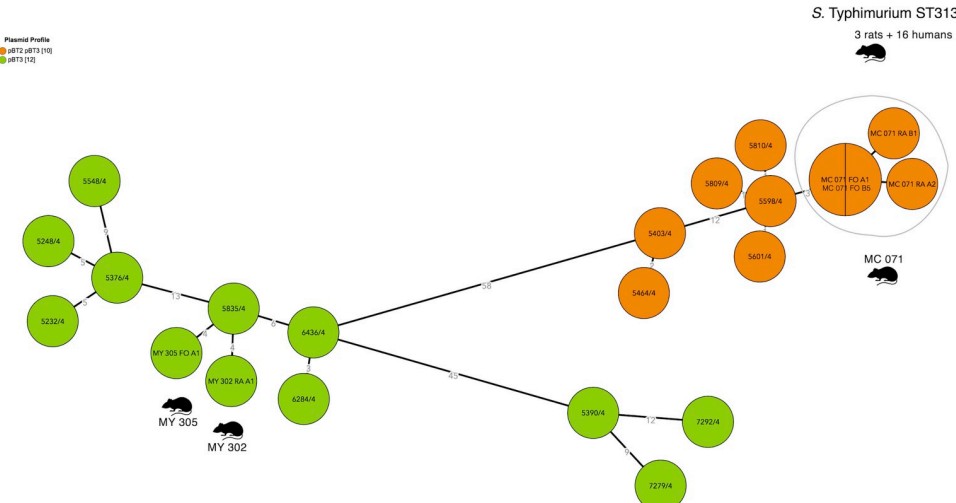

**Fig 4. Plasmid profiles of *S.* Typhimurium ST313 human and rat isolates according to the plasmid profiles of ST313 Lineage 2 isolates described by Pulford *et al.* [7].** Minimum spanning tree (MSTree V2) comparing all *S.* Typhimurium ST313 in this study. 58 alleles separate the two mixed clusters (human and rats isolates) with different plasmid content: in orange, isolates harboring plasmids pBT2 and pBT3; in green, isolates harboring pBT3, only. Rat drawing by Francisca Arévalo from NounProject.com.

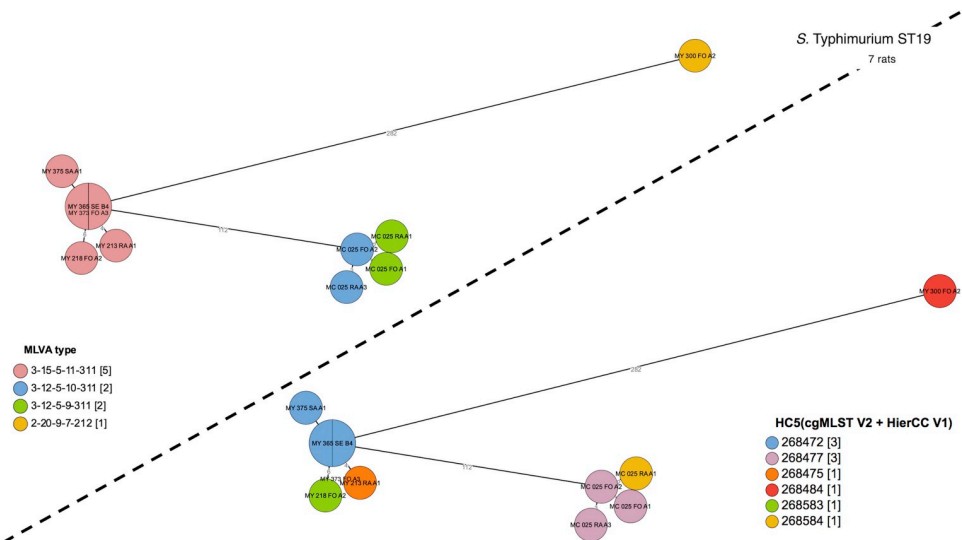

**Fig 5. Clustering of *S.* Typhimurium ST19 rat isolates according to HC5-values (cgMLST V2 + HierCC V1, upper left) and MLVA type (lower right).** Minimum spanning tree (MSTree V2) comparing all ST19 in this study. In panel A, colors are according to cgMLST HC5 values; in panel B, colors are according to MLVA type.

layer or pig farms (up to 28%) [29–32,46–50]. Of note, studies also showed a high variation in *Salmonella* prevalence between rats in different cities; 15% in Yokohama (Japan) versus 1% in Chicago (US), 0.34% in Nairobi (Kenya) and 0% in Baghdad (Iraq) [32,51–53]. These differences can partly be ascribed to the urban landscape (presence of food and shelter, sanitary conditions); the highest reported prevalence (49.1%) has been observed in wet marketplaces in Udon Thani City, Thailand [54].

In the above-mentioned studies, *Salmonella* serotypes from commensal rats varied widely but there was a consistent overlap with human epidemiologically relevant serotypes. Examples are *Salmonella* Enteritidis and Infantis from layer farms [26,47,55] and *Salmonella* Weltevreden from rats in marketplaces in Thailand [54]. *S.* Typhimurium and–to a lesser extent–Enteritidis belong to the most observed serotypes [29,31,32,47,48,52,54,56]. Molecular typing allowing for sequence typing of the non-Typhoidal *Salmonella* was not performed so far. The aforementioned study in Japan showed that *S.* Typhimurium isolates obtained from urban rats

**Table 4. Antimicrobial resistance profiles of *Salmonella* Typhimurium isolates from rats and humans (blood cultures).** Data present numbers (%) of resistant isolates. Only the first isolate per serotype and rat is listed. MDR: multidrug resistant, *i.e.* resistant to ampicillin, trimethoprim-sulfamethoxazole and chloramphenicol. DCS = decreased ciprofloxacin susceptibility, *i.e.* ciprofloxacin MIC-value > 0.064 mg/L and < 1 mg/L.

| Resistance to: | Rats | | | Humans |
|---|---|---|---|---|
| | *Salmonella* Typhimurium ST 313 (N = 3) | *Salmonella* Typhimurium ST 19 (N = 7) | Other *Salmonella* (N = 52) | *Salmonella* Typhimurium (N = 18) |
| Ampicillin | 3 | 0 | 0 | 18 |
| Trimethoprim-sulfamethoxazole | 3 | 0 | 0 | 18 |
| Chloramphenicol | 3 | 0 | 0 | 14 |
| Multidrug resistant (MDR) | 3 (100.0%) | 0 (0.0%) | 0 (0.0%) | 14 (77.8%) |
| Ceftriaxone | 0 | 0 | 0 | 1 |
| Azithromycin | 0 | 0 | 0 | 4 |
| Decreased ciprofloxacin susceptibility (DCS) | 0 | 0 | 0 | 3 |
| MDR + DCS | 0 | 0 | 0 | 1 |

belonged to the DT104 phage type, *i.e.*, the world-wide distributed zoonotic *S.* Typhimurium causing diarrhea in humans [32].

## 4.2 Rats carried *S.* Typhimurium ST313, the predominant *S.* Typhimurium clone in Africa

The 8.1% *Salmonella* prevalence among rats in the present study fits into the prevalence ranges described above. Among the serotypes recovered, the main finding was the presence of *S.* Typhimurium ST313 in three rats. By MLVA as well as by comparison of genomes in Entero-Base, the ST313 isolates grouped in two clusters comprising each six human ST313 isolates recovered from blood cultures. ST313 is the main pathovariant of iNTS in Africa [7]. Both rat and human isolates clustered together with what Pulford *et al.* describe as Lineage 2, *i.e.* the most common and recent *S.* Typhimurium ST313 lineage in the African continent [7]. Close genetic relation was also supported by the identification of the pSLT-BT plasmid, encoding for the different AMR genes that most ST313 isolates from rats and humans in the present study harbored, particularly the *cat1* gene encoding for chloramphenicol resistance, which is hallmark for Lineage 2 [7].

## 4.3 Finding of *Salmonella* Typhimurium ST313 in rats: laboratory animal models

So far, reservoir and transmission of iNTS including ST313 were hypothesized to be human, based on the genetic signature [4,5,22] and household, case-control, and environmental studies [15,16,20].

Finding ST313 in rats indicates that ST313 is not restricted to the human host. Previous laboratory animal models showed that ST313 was able to cause invasive infections in chickens, mice, and rhesus macaque infection models [6,57–59]. Findings differed according to the challenging strain and host animal but overall ST313 caused, in comparison with ST19, a more rapid onset of bacteremia and infestation of liver and spleen combined with less diarrhea and colonization of the intestinal tract. The fact that in the present study ST313 in rats was exclusively recovered from deep organs may point to an invasive disease too, but none of the rats captured was visibly sick and their length and mass were in line with the other rats. Moreover, although *S.* Typhimurium strains can cause deadly infections, asymptomatic infections of liver and spleen with intermittent fecal shedding can occur too [29,60,61] and may even constitute the majority of infections [62].

## 4.4 Attribution of rats to Salmonella transmission

Despite being acknowledged as a reservoir for *Salmonella*, there is only circumstantial and anecdotal evidence as to the attribution of rats to the transmission of *Salmonella* [29,61,63]. Rather than being an original reservoir for *Salmonella*, rats are considered to be a "sponge" reflecting the environment they are living in [26,45,51]. This has been demonstrated in layer farms, where proportions of *Salmonella* carrying rats were related to the percentage of *Salmonella* contaminated eggs and the intensity of the environmental contamination [47].

Rats shed *Salmonella* for 2–4 months after infection and have up to 40 droppings a day [29]; *Salmonella* survives long (up to 86 days) in rat droppings [30]. In this way, rats contaminate the layer hen's food and thereby maintain and even amplify *Salmonella* infection in the food production chain [26,47,55]. Conducive to this is the social life of rats (allowing transmission among rats [64] and the long (> 1 year) environmental survival of *Salmonella* [65], although ST313 may have lower environmental persistence compared to the diarrhea-causing

ST19 [11]. In addition, rat carcasses may also constitute a *Salmonella* reservoir on which other animals (stray cats, lizards, cockroaches, ants, etc.) can feed and further contribute to the spread of *Salmonella* [26].

A similar mechanism of indirect ("sponge") reservoir may be postulated for wet markets. In the present study, this is supported by the observation that the marketplaces where ST313 carrying rats were captured (Central Market and Yaté market) were those with the highest rat density. Further, the market setting in a low resource setting such as Kisangani offers exposure of food and shelter in the absence of pest control. By consequence, these are high density populations of rats which was demonstrated by the high trapping success in the present study (one third of non-pre-baited traps) [45]. The present rat population consisted of nearly 90% sexually mature rats versus 63.9% in a rat population in urban Canada; also, length and mass were higher in Kisangani versus Canada (medians of 22.1 cm and 280 g versus 17.5 cm and 142.2 g respectively). This means that the Kisangani rats were probably older, which may increase the probability of rat associated zoonosis [45].

At the human side, socioeconomic factors fueling transmission in Kisangani are numerous: poor sanitary conditions, crowding, overnight stay of vendors, unsafe drinking water and cooking practices. The marketplaces sampled in the present study are easy to reach by the Kisangani population, either by walking or by motorcycle, and human mobility is compatible with the geographical perimeters of the rat-human clusters in the present study. Of note, rats can move too: urban rats are territorial [32,45] but may walk up to 3 km and further in one night [66,67]

## 4.5 The present findings point to the evidence of an animal reservoir for ST313

As only 6.5% of *Salmonella* carrying rats and 0.5% of all rats captured harbored ST313, it is unlikely that rats constitute a major or preferred reservoir of ST313. However, the rat ST313 isolates clustered with the majority (75.0%) of human isolates, pointing to a close interaction between both. Moreover, the occurrence of ST313 in rat populations postulates also other animals as potential reservoirs of ST313. Previous household studies assessing the iNTS reservoirs in sub-Saharan Africa sampled livestock and poultry but no indoor rodents [15,20,68]. Recently, two studies assessed the meat pathway as a potential source of *Salmonella*. A slaughterhouse study from Kenya and Malawi found a 12.7% *Salmonella* prevalence in pig carcasses but failed to detect iNTS clades [66]. A study from East Africa showed that *S*. Enteritidis ST11 was present in the meat pathway and clustered with human isolates. Other non-typhoidal *Salmonella* serotypes were observed in the meat pathway too, but not ST313 [23]. Furthermore, a study in DR Congo showed a direct association of rainfall with iNTS, pointing to a possible waterborne environmental reservoir [69].

The potential existence of an environmental reservoir and transmission of iNTS may have an impact on ongoing efforts on iNTS vaccine development, most of which target *S*. Typhimurium [70] and put more emphasis on water, sanitation, and hygiene as control measures. Moreover, ongoing deforestation and climate change are expected to favor both rodent populations and salmonellosis [65,71] with a potential increasing impact of the rat-*Salmonella* reservoir and transmission.

## 4.6 Other Salmonella serotypes and antimicrobial resistance

The other serotypes obtained from rats mainly comprised II:42:r:-, Kapemba, Weltevreden and Dublin. They were susceptible to all antibiotics tested (= pan-susceptible), as was also the case for the *S*. Typhimurium ST19 isolates. This pan-susceptibility might be explained by the

fact that the non-ST313 *Salmonella* isolates most probably represent animal-confined isolates. Agriculture and livestock raising in the Kisangani area are mainly organized as small-scale subsistence farming with expected a lesser use of antibiotics compared to human medicine. A similar observation was noted in the aforementioned study about *Salmonella* in pigs in Kenya and Malawi: most isolates were susceptible to all antibiotics tested, in line with presumed low antibiotic exposure [72].

*Salmonella* Dublin is host-adapted to cattle but has been recognized as iNTS too in Mali [73]; it has not yet been detected as part of the microbiological surveillance in DR Congo [22,71]. *Salmonella* Weltevreden is a major cause of intestinal infections in Asia and was cultured from rats in marketplaces in Thailand and India in different environments as poultry farm, feed store and around residential quarters [48,54]. To our knowledge, *Salmonella* serotypes II:42:r:-, Kapemba, Mikawasima and Orion so far have not yet been described in rats.

## 4.7 Limitations and strengths

A limitation of the present study was the fact that only outdoor sampling was conducted. Indoor sampling would have been a valuable adjunct but in view of the public perception and acceptability, it was presently not done. Likewise, traps were not pre-baited whereas doing so could have increased trapping success as rats are neophobic [45,67]. Trapping success was however high and in line with a previous study [25]. Last, the molecular subtyping alone (as presently used) cannot provide information about the pathways of transmission (rat to human or vice-versa) [13].

Among the strengths were the multiple sampling sites which precluded rat territorial bias [74]. Further, trapping was not impacted by non-target species and, thanks to excellent communication with the city and market authorities, there was no tripping or stealing by the public [74]. The present sample size (n = 566) was large compared to other studies [48,51,72] and human clinical isolates from the same period and area were available for comparison. In addition, we collected physical data of the rats (which may relate to the presence of zoonosis [74] and sampled year-round, precluding seasonal variation. The co-located laboratory facilities allowed for expertized staff and short and regular transport. Deep organs were sampled, and they accounted for three quarters of the non-duplicate serotype isolates. If only stool/rectal swab or content would have been sampled (as done in many studies [32,45,54,75]), carriage would have been as low as 2.6% (15 rats) and ST313 would have been missed. Furthermore, as confirmed here, rats may carry more than one *Salmonella* serotypes [44,45], which is the reason why we processed multiple colonies per sample.

Further research should confirm and extend the present findings to other iNTS-endemic areas. Field sampling should include wet marketplaces but also indoor rodent sampling in households of iNTS index patients. In conjunction with rat studies, market retail products should be assessed for contamination with iNTS [76]. Laboratory experiments should establish the pathophysiology of iNTS in rats in particular intestinal colonization and fecal shedding. At a larger scope, further environmental research into the reservoir of iNTS should be conducted [6,21,23]. Apart from rats, small rodents such as mice should be assessed, as well as lizards and geckos which are notable carriers of *Salmonella* too [77]. Food and water should be further assessed as a vehicle for transmission, including fork-to-farm studies along the meat pathway [23,78], crops and green leaves [13,69]. Source attribution studies such as case-control and outbreak investigation studies should complement the picture of iNTS transmission [13,78].

In conclusion, the present study provided evidence of rats as carriers of *S.* Typhimurium ST313 clustering with human blood culture isolates in an iNTS endemic area in sub-Saharan Africa.

## Supporting information

**S1 Table. ENA depository genomes.**
(DOCX)

**S2 Table. Morphometric parameters of rats (*Rattus rattus* and *Rattus norvegicus*) related to the *Salmonella* carriage status in Kisangani 2016–2018).**
(DOCX)

**S3 Table. Antibiotic susceptibility data of human and rat *Salmonella* isolates.**
(DOCX)

**S1 Fig. Timeline of human and rat sampling.**
(DOCX)

**S2 Fig. Clustering of human *Salmonella* Enteritidis isolates with ST11 *Salmonella* Enteritidis isolates.**
(DOCX)

## Acknowledgments

We would like to thank the technical staff participating in the fieldwork, dissection, and laboratory analysis, in particular Brigitte Mapendo and Rémy Bongolu.

## Author Contributions

**Conceptualization:** Dadi Falay, Chris Van Geet, Dauly Ngbonda, Jan Jacobs.

**Data curation:** Dadi Falay, Liselotte Hardy, Octavie Lunguya, Maria Pardos de la Gandara, Jan Jacobs.

**Formal analysis:** Dadi Falay, Maria Pardos de la Gandara, Jan Jacobs.

**Funding acquisition:** Chris Van Geet, Dauly Ngbonda, Jan Jacobs.

**Investigation:** Dadi Falay, Wesley Mattheus, Pionus Katuala, Maria Pardos de la Gandara, Jan Jacobs.

**Project administration:** Dadi Falay, Jan Jacobs.

**Resources:** Chris Van Geet, Pionus Katuala, Dauly Ngbonda, François-Xavier Weill.

**Supervision:** Liselotte Hardy, Octavie Lunguya, Chris Van Geet, Erik Verheyen, Dauly Ngbonda, François-Xavier Weill, Jan Jacobs.

**Visualization:** Dadi Falay, Liselotte Hardy, Maria Pardos de la Gandara, Jan Jacobs.

**Writing – original draft:** Dadi Falay, Maria Pardos de la Gandara, Jan Jacobs.

**Writing – review & editing:** Liselotte Hardy, Jacques Tanzito, Octavie Lunguya, Edmonde Bonebe, Marjan Peeters, Wesley Mattheus, Chris Van Geet, Erik Verheyen, Dudu Akaibe, Pionus Katuala, Dauly Ngbonda, François-Xavier Weill, Maria Pardos de la Gandara.

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
