## [Decision Letter · Decision Letter 0]

21 Jun 2022

Dear Dr. Hardy,

Thank you very much for submitting your manuscript "Urban rats as carriers of invasive Salmonella Typhimurium sequence type 313, Kisangani, Democratic Republic of Congo." for consideration at PLOS Neglected Tropical Diseases. As with all papers reviewed by the journal, your manuscript was reviewed by members of the editorial board and by several independent reviewers. The reviewers appreciated the attention to an important topic. Based on the reviews, we are likely to accept this manuscript for publication, providing that you modify the manuscript according to the review recommendations. 

Please note editorial comments made by reviewer 1, along with suggestions for more detail on the rodent trapping methods used and defining “trapping success” in the methods section of the revised manuscript. 

Both reviewers 1 & 3 asked for a map with rodent trapping sites indicated. 

Reviewer 3 also requested an excel file to include data used from other groups/public databases. A separate excel file with detailed MIC data should also be provided with the revised manuscript.

All genomic data used in this study will need to be made available per PLoS NTDs guidelines.

Sincerely,

Travis J Bourret

Associate Editor

Alfredo Torres

Deputy Editor

Please note editorial comments made by reviewer 1, along with suggestions for more detail on the rodent trapping methods used and defining “trapping success” in the methods section of the revised manuscript. 

Both reviewers 1 & 3 asked for a map with rodent trapping sites indicated. 

Reviewer 3 also requested an excel file to include data used from other groups/public databases. A separate excel file with detailed MIC data should also be provided with the revised manuscript.

All genomic data used in this study will need to be made available per PLoS NTDs guidelines.

Reviewer's Responses to Questions

**Key Review Criteria Required for Acceptance?**

**Methods**

-Are the objectives of the study clearly articulated with a clear testable hypothesis stated?

-Is the study design appropriate to address the stated objectives?

-Is the population clearly described and appropriate for the hypothesis being tested?

-Is the sample size sufficient to ensure adequate power to address the hypothesis being tested?

-Were correct statistical analysis used to support conclusions?

-Are there concerns about ethical or regulatory requirements being met?

Reviewer #1: -Are the objectives of the study clearly articulated with a clear testable hypothesis stated?

Yes.

-Is the study design appropriate to address the stated objectives?

Yes.

-Is the population clearly described and appropriate for the hypothesis being tested?

Yes.

-Is the sample size sufficient to ensure adequate power to address the hypothesis being tested?

Yes.

-Were correct statistical analysis used to support conclusions?

Yes.

-Are there concerns about ethical or regulatory requirements being met?

No.

Other comments:

1. Sometimes nontyphoidal Salmonella (NTS) is qualified with ‘invasive’ unnecessarily or inappropriately. For example, as a redundancy when bloodstream isolates are being described, and perhaps incorrectly when carriage isolates are being described (i.e., isolates from rectal content, when a definition for invasion is not met).

2. The term ‘pan-susceptible’ is used without definition. Relatedly, a full listing of antimicrobial agents tested is not provided in the Methods.

3. There seems to be some conflation of carriage, infection, and disease terms in relation to rat isolates, with 'carriage' being used when infection or disease is not comprehensively excluded.

4. Please check all occurrences of the word ‘rate.’ On several occasions the word ‘rate’ is used to describe a proportion or prevalence, rather than a time-denominated measure of incidence. For example, ‘case fatality rate’ is used when ‘case fatality ratio’ would be more appropriate, and ‘high rate of antimicrobial resistance’ is used when ‘high prevalence of antimicrobial resistance’ is intended. A lot has been written about this (pedantic) point, but one recent example is by S Mantha 'Ratio, rate, or risk?' Lancet Infect Dis 2020; 21: 165-6.

5. Oxford (serial) commas would be helpful for clarity in places.

6. Nontyphoidal Salmonella cause in humans, anatomically and pathologically, predominantly entercolitis rather than gastroenteritis. If you wish to be anatomically specific, I would switch terms to enterocolitis. If you wish to avoid this issue, perhaps a clinical term like ‘diarrhea’ would be more accurate.

7. ‘Paratyphi’ is not a recognized serovar of Salmonella enterica, but Salmonella Paratyphi A, Paratyphi B, and Paratyphi C are.

8. It would be useful to say a little more about non-genomic evidence for NTS adaptation to humans in the Introduction.

9. A little more detail on rodent trapping methods would be helpful. For example, site selection process, numbers of traps set per night per site, dates of trapping (in relation to seasons), whether trapping was outdoor, or indoor, or both, etc.

10. What was the median (range) time in hours (or target) between trapping and euthanasia (i.e., lairage time)? What was the median (range) time in hours (or target) between euthanasia and necropsy? What storage conditions were rodents kept in after euthanasia and before necropsy? These points are relevant to onset of recent bacteremia in rats in prolonged lairage, and for postmortem translocation of organism from the gut to deep tissues.

11. Little description of how ‘signs of illness’ in rats was assessed, nor of gross findings at rat necropsy are provided. Were gross findings besides length and weight recorded? I assume no histopathology of liver and spleen to rule out signs of inflammation?

12. What was the rationale for selecting up to five colonies, rather than a higher or lower number?

13. A definition of a ‘carrier’ would be helpful. The case for calling isolation from spleen and liver ‘carriage’ is made, but I am concerned that they could represent disease isolates since histopathologic studies were not done, and details of gross necropsy findings were scanty in the report.

14. The term ‘antibiotic’ is used when ‘antimicrobial’ may be more appropriate.

Reviewer #2: The methods are adequate for the study question.

Reviewer #3: yes, technically sound with a clear aim.

Line 107-108, how the sample calculation was conducted?

Detailed excel sheet regarding the data used (from other groups or public datasets) are needed.

**Results**

-Does the analysis presented match the analysis plan?

-Are the results clearly and completely presented?

-Are the figures (Tables, Images) of sufficient quality for clarity?

Reviewer #1: -Does the analysis presented match the analysis plan?

Yes.

-Are the results clearly and completely presented?

Yes.

-Are the figures (Tables, Images) of sufficient quality for clarity?

Yes.

Other comments:

1. I suggest that numerators, proportions, and denominators be presented consistently in the conventional manner i.e., ‘x (y%) of z,’ or ‘Of z, x (y%)…’ Always give actual numbers for numerators and denominators. For example, I would avoid ‘one third’ as a proxy numerator.

2. ‘Trapping success’ is not defined in the Methods. Is this per trap, per site, or something else?

3. A map of greater Kisangani with trapping sites would be helpful. Adding relevant human participants would also be useful if geolocated and ethically acceptable.

Reviewer #2: The results are adequately describing the study outcomes.

Reviewer #3: An excel sheet with detailed samples as well as their meta data is needed.

Is there any rat samples having multiple Salmonella isolates or serovars?

Detailed excel sheet regarding the MIC data should also be included.

Further serovar based analysis regarding Salmonella Dublin, II:42:r:-, Kapemba, Weltevreden is needed, some are important and potential hazards in Africa countries and many others.

A map of sampling is needed.

All genomic data is not available.

**Conclusions**

-Are the conclusions supported by the data presented?

-Are the limitations of analysis clearly described?

-Do the authors discuss how these data can be helpful to advance our understanding of the topic under study?

-Is public health relevance addressed?

Reviewer #1: -Are the conclusions supported by the data presented?

Yes.

-Are the limitations of analysis clearly described?

Yes.

-Do the authors discuss how these data can be helpful to advance our understanding of the topic under study?

Yes.

-Is public health relevance addressed?

Yes.

Other comments:

1. Since Salmonella Weltevreden was a common serovar isolated from rats, it would be useful to know whether the markets to which rodents had access in Kisangani sold fish, including farmed fish.

2. More discussion on the interpretation of isolation of Salmonella from liver and spleen would be helpful. In particular, could this represent recent onset bacteremia following prolonged lairage, or post-mortem translocation from the gut to the deep tissues?

3. The term ‘rat-to-rat’ transmission but be interpreted by some readers with an epidemiologic background as an indication of transmission by direct contact (c.f., fecally contaminated water or food, feces consumption, etc). I assume that is not the authors' intent, and wonder if ‘transmission among rats’ may be less likely to imply insights into the mode of transmission.

4. It is likely going beyond the scope of the findings, but would the authors care to speculate on possible modes of transmission from rodents to people, should rodents be a reservoir here? For example, we are not told whether human slaughter and consumption of rats and other rodents is practiced in Kisangani. I assume that contamination of water and food by rodent feces is likely, but not explicitly mentioned.

Reviewer #2: The conclusions are adequately describing the study outcomes.

Reviewer #3: Rats as reservoirs of non-Typhoidal Salmonella, and many more references should be included.

The references in the field should be updated.

**Editorial and Data Presentation Modifications?**

Reviewer #1: Please see suggested modifications under Methods, Results, and Conclusions.

Reviewer #2: (No Response)

Reviewer #3: Minor revision

**Summary and General Comments**

Reviewer #1: This is an extremely important study that sheds light on rats as possible non-human reservoirs for nontyphoidal Salmonella strains causing invasive disease in humans in Africa. I have made a number of relatively minor points for improvement outlined in my comments by section.

Reviewer #2: The study sheds some light on the transmission patterns of iNTS disease and it will be important to publish these results.

Reviewer #3: Dadu Falay and colleagues (PNTD-D-22-00687) presented a large scale investigation regarding the Salmonella carriage in Urban rats. The study sampled all positive Salmonella, which was subjected into WGS, and compared with the human isolates. The overall data is very interesting and could guide iNTS infection in Africa.

PLOS authors have the option to publish the peer review history of their article (what does this mean?). If published, this will include your full peer review and any attached files.

Reviewer #1: No

Reviewer #2: No

Reviewer #3: No

Figure Files:

Data Requirements:

Reproducibility:

References

---

## [Decision Letter · Decision Letter 1]

13 Aug 2022

Dear Dr. Hardy,

We are pleased to inform you that your manuscript 'Urban rats as carriers of invasive Salmonella Typhimurium sequence type 313, Kisangani, Democratic Republic of Congo.' has been provisionally accepted for publication in PLOS Neglected Tropical Diseases.

Best regards,

Travis J Bourret

Academic Editor

Alfredo Torres

Section Editor

Reviewer's Responses to Questions

**Key Review Criteria Required for Acceptance?**

**Methods**

-Are the objectives of the study clearly articulated with a clear testable hypothesis stated?

-Is the study design appropriate to address the stated objectives?

-Is the population clearly described and appropriate for the hypothesis being tested?

-Is the sample size sufficient to ensure adequate power to address the hypothesis being tested?

-Were correct statistical analysis used to support conclusions?

-Are there concerns about ethical or regulatory requirements being met?

Reviewer #1: My only remaining concern is with the use of the term 'carriage' to refer to infection of (or isolation of a pathogen from) a normally sterile site like liver, spleen, or blood. While the authors can define terms however they like, I think that this use of 'carriage' is beyond what most readers would consider conventional.

Reviewer #3: (No Response)

**Results**

-Does the analysis presented match the analysis plan?

-Are the results clearly and completely presented?

-Are the figures (Tables, Images) of sufficient quality for clarity?

Reviewer #1: My only remaining concern is with the use of the term 'carriage' to refer to infection of (or isolation of a pathogen from) a normally sterile site like liver, spleen, or blood. While the authors can define terms however they like, I think that this use of 'carriage' is beyond what most readers would consider conventional.

Reviewer #3: (No Response)

**Conclusions**

-Are the conclusions supported by the data presented?

-Are the limitations of analysis clearly described?

-Do the authors discuss how these data can be helpful to advance our understanding of the topic under study?

-Is public health relevance addressed?

Reviewer #1: Yes.

Reviewer #3: (No Response)

**Editorial and Data Presentation Modifications?**

Reviewer #1: None.

Reviewer #3: (No Response)

**Summary and General Comments**

Reviewer #1: The authors have done a very nice job of responding to my comments. My only remaining concern is with the use of the term 'carriage' to refer to infection of (or isolation of a pathogen from) a normally sterile site like liver, spleen, or blood. While the authors can define terms however they like, I think that this use of 'carriage' is beyond what most readers would consider conventional.

Reviewer #3: (No Response)

PLOS authors have the option to publish the peer review history of their article (what does this mean?). If published, this will include your full peer review and any attached files.

Reviewer #1: No

Reviewer #3: **Yes: **Min Yue

---

## [Editor Report · Acceptance letter]

30 Aug 2022

Dear Dr Hardy,

We are delighted to inform you that your manuscript, "Urban rats as carriers of invasive *Salmonella* Typhimurium sequence type 313, Kisangani, Democratic Republic of Congo.," has been formally accepted for publication in PLOS Neglected Tropical Diseases.

Best regards,

Shaden Kamhawi

co-Editor-in-Chief

Paul Brindley

co-Editor-in-Chief
